# Ergosterol Isolated from *Antrodia camphorata* Suppresses LPS-Induced Neuroinflammatory Responses in Microglia Cells and ICR Mice

**DOI:** 10.3390/molecules28052406

**Published:** 2023-03-06

**Authors:** Ping Sun, Weiling Li, Jiazheng Guo, Qian Peng, Xiansheng Ye, Song Hu, Yuchen Liu, Wei Liu, Haifeng Chen, Jialu Qiao, Binlian Sun

**Affiliations:** 1Wuhan Institute of Biomedical Sciences, School of Medicine, Jianghan University, Wuhan 430056, China; sinpisces@sina.com (P.S.); liwl.whibs@jhun.edu.cn (W.L.); g1053074710@163.com (J.G.); pengqian199034@163.com (Q.P.); yxs178189@163.com (X.Y.); husong@jhun.edu.cn (S.H.); yuchen.liu@jhun.edu.cn (Y.L.); liuwei@jhun.edu.cn (W.L.); jialuqiao@jhun.edu.cn (J.Q.); 2Fujian Provincial Key Laboratory of Innovative Drug Target, School of Pharmaceutical Sciences, Xiamen University, Xiamen 361005, China; haifeng@xmu.edu.cn

**Keywords:** LPS, neuroinflammation, microglia, Ergosterol, NF-κB

## Abstract

Inflammation caused by microglial activation is important in neurodegenerative diseases. In this research, we tried to identify safe and effective anti-neuroinflammatory agents by screening a natural compounds library and found that Ergosterol can inhibit the nuclear factor kappa-light-chain enhancer of the activated B cells (NF-κB) pathway induced by lipopolysaccharide (LPS) in microglia cells. Ergosterol has been reported to be an effective anti-inflammatory agent. Nevertheless, the potential regulatory role of Ergosterol in neuroinflammatory responses has not been fully investigated. We further investigated the mechanism of Ergosterol that regulates LPS-induced microglial activation and neuroinflammatory reactions both in vitro and in vivo. The results showed that Ergosterol can significantly decrease the pro-inflammatory cytokines induced by LPS in BV2 and HMC3 microglial cells, possibly by inhibiting the NF-κB, protein kinase B (AKT), and mitogen-activated protein kinase (MAPK) signaling pathways. In addition, we treated Institute of Cancer Research (ICR) mice with a safe concentration of Ergosterol following LPS injection. Ergosterol treatment significantly decreased microglial activation–associated ionized calcium-binding adapter molecule-1 (IBA-1), NF-κB phosphorylation, and pro-inflammatory cytokine levels. Moreover, Ergosterol pretreatment clearly reduced LPS-induced neuron damage by restoring the expression of synaptic proteins. Our data may provide insight into possible therapeutic strategies for neuroinflammatory disorders.

## 1. Introduction

Inflammation is a major contributor to many neurodegenerative disorders, including Alzheimer’s disease (AD), Parkinson’s disease (PD), amyotrophic lateral sclerosis (ALS), multiple sclerosis (MS), and an increasing number of other diseases of the nervous system [1,2]. The immune system has an important role in maintaining tissue homeostasis and the response to infection and injury. Microglia, a type of glial cell, is a special type of macrophage that resides in the brain. They constantly monitor the microenvironment and produce factors that affect the surrounding astrocytes and neurons. However, after chronic activation, microglia become dysfunctional, characterized by morphological changes and elevated amounts of inflammatory cytokines and chemokines, leading to neuronal instability and degeneration [3,4]. Hence, inhibition of the activation of microglia is a potential mechanism for preventing the development of neuroinflammation and related diseases.

To alleviate neuroinflammation, antioxidant compounds of plant origin are considered safe for use [5]. Natural substances from plants have been found to provide neuroprotective activity. Ginsenoside Rg3 effectively reduces the expression of inflammatory cytokines by suppressing nuclear factor kappa-light-chain enhancer of activated B cell P65 (NF-κB P65), binding to its DNA consensus sequence in activated microglia [6]. Resveratrol, generally found in grapevines and other fruits, has shown suppression in inflammatory responses and a decrease in amyloid β-protein (Aβ) levels, resulting in the amelioration of cognitive function [7]. Epigallocatechin gallate (EGCG), the most abundant polyphenol component in green tea, was found to decrease cognitive impairment in β-amyloid precursor protein/presenilin 1 (APP/PS1) mice by increasing nerve growth factor (NGF) expression and by promoting Cyclic Adenosine monophosphate (cAMP)-response element binding protein (CREB) expression [8]. Therefore, developing safe and effective neuroprotective agents from phytochemicals is a possible method for treating neurological disorders caused by neuroinflammation.

Lipopolysaccharide (LPS) is an inducer that activates microglia through interaction, and it is identified by the toll-like receptor 4 (TLR4) located on the surface of microglial cells [9]. Hence, LPS activates the signaling pathways downstream of TLR4, including the mitogen-activated protein kinase (MAPK) and protein kinase B (AKT) signaling pathways [10,11], along with NF-κB, bringing about the enhanced activation of pro-neuroinflammatory mediators, resulting in higher cytokine expression and release [12].

Ergosterol is the precursor of steroid hormone and vitamin D2 in the pharmaceutical industry [13], and it is a secondary metabolite common to various fungi, with beneficial health effects such as anti-inflammatory, anti-tumor, and anti-viral activity, as well as reducing the incidence of cardiovascular disease [14]. It has been reported that Ergosterol blocks the NF-κB signaling pathway in acute lung injury, which is caused by LPS in human bronchial epidermal cells (16HBE) and mouse lungs [15]. Another study showed that Ergosterol is an anti-inflammatory agent by restraining the production of tumor necrosis factor-α (TNF-α) and the expression of cyclooxygenase-2 (COX-2) in macrophages of RAW 264.7 and by reducing nitric oxide (NO) in BV2 microglia [16,17]. These findings suggest that Ergosterol might have anti-inflammatory effects by interfering with several pathways. However, the potential for Ergosterol to regulate brain neuroinflammatory responses has not been fully investigated.

In this study, our results showed that Ergosterol suppresses the mRNA levels of interleukin-1β (IL-1β) and TNF-α in both BV2 and HMC3 microglial cell lines treated with LPS by inhibiting the NF-κB, MAPK, and AKT signaling pathways. Further investigation found that Ergosterol treatment significantly decreased the microgliosis-linked associated ionized calcium-binding adapter molecule-1(IBA-1) level, which is generally considered a marker of microglial activation [18], NF-κB phosphorylation, and proinflammatory cytokines, and it clearly reduced neuron damage by restoring the expression of synaptic proteins in LPS-injected ICR mice. Taken together, our findings suggest that Ergosterol could suppress the activation of the microglia and neuroinflammatory responses induced by LPS both in vitro and in vivo via inhibiting multi-pathways.

## 2. Results

### 2.1. Ergosterol Inhibits LPS-Induced Microglial Inflammation

To screen anti-neuroinflammation natural compounds, BV2 cells, a microglial cell line from mice, were preincubated with a 50 μM compound library containing 49 natural products. After 2.5 h, the cells were treated with LPS for 30 min to produce an inflammation model. The cells were then collected, and phospho-NF-κB P65 (p-P65) was detected in the cell lysates. The results showed that LPS treatment could increase the phosphorylation level of P65 to activate the NF-κB pathway, and four compounds (Nos. 5, 24, 48, 49) suppressed this response. We further detected the anti-inflammation of these four compounds by detecting several inflammatory mediator proteins in the BV2 cells. It transpired that LPS treatment increased the phosphorylation level of the multiple inflammatory mediators, including p-P65, phosphorylated P38 (p-P38), phospho-JNK (p-JNK), and phospho-AKT (p-AKT), which again indicated that the LPS-induced inflammation model was successful in the BV2 cells and that the four compounds efficiently suppressed the increase in these mediators induced by LPS, with No. 5 showing a stronger function (Figure 1A). Next, the cytokine expression level, downstream of the inflammatory pathway, was detected with a qRT-PCR assay. The results showed that these compounds suppressed the mRNA levels of IL-1β, TNF-α, and interleukin-18 (IL-18) in the BV2 cells treated with LPS (Figure 1B–D). In addition, No. 5 exhibited the most inhibition activity with regard to cytokine expression. Furthermore, we confirmed the function of No. 5 in human microglial cell line HMC3, with qRT-PCR detection showing that LPS could also induce intense inflammation in the HMC3 cells, while No. 5 also suppressed the expression of IL-1β and TNF-α, but did not affect the expression of IL-18 (Figure 1E). The above results implied that the No. 5 compound has an anti-inflammatory function by suppressing several signaling pathways.

As the markers of microglial cell activation and pro-inflammatory mediators, inducible nitric oxide synthase (INOS) and cyclooxygenase 2 (COX-2) expression levels were also detected. The results showed that No. 5 also decreased the expression levels of COX-2 and INOS induced by LPS in the BV2 and HMC3 cells in a compound dose-dependent manner (Figure 1F,G), suggesting that the No. 5 compound can inhibit the activity of microglial cells via suppressing the expression of inflammatory mediators. We then identified the ingredient of No. 5 using nuclear magnetic resonance (NMR) analysis and high resolution electrospray ionization mass spectroscopy (HRESIMS), finding that it was Ergosterol (Appendix A and Figure 1H), which is reported to have an anti-inflammatory function [19,20].

Thereafter, we confirmed the potential cytotoxicity of Ergosterol using a Cell Counting Kit-8 (CCK8) assay. The BV2 and HMC3 cells were treated with Ergosterol (5, 10, 30, 50, or 100 μM) for 24 h, and CCK8 detection indicated that Ergosterol had no effect on cell viability at doses within the range of 5 to 50 μM (Figure 1I), indicating that Ergosterol is safe in both BV2 and HMC3 cells under 50 μM. Taken together, the above data suggest that Ergosterol has good inhibition activity regarding LPS-induced microglial inflammation without cytotoxicity.

### 2.2. Ergosterol Suppresses NF-κB Signaling Activated by LPS in BV2 and HMC3 Microglial Cells

The increase in P65 phosphorylation is a sign of NF-κB signaling pathway activation, which is the main transcription factor for cytokine expression. We went a step further to investigate the influence of Ergosterol acting on the NF-κB pathway. Western blotting showed that the p-P65 increase induced by LPS was reduced in an Ergosterol concentration-dependent manner in BV2 cells (Figure 2A,B). At the same time, pretreatment with Ergosterol significantly inhibited LPS’s induction of IL-1β mRNA expression in the BV2 cells (Figure 2E). In addition, the same experiments were performed in HMC3 cells, with the p-P65 protein and IL-1β mRNA detection showing similar results (Figure 2C,D,F). The nuclear translocation of NF-κB is a precondition for activation. To further confirm the above results, the expression and location of another NF-κB subunit, P50, were detected by immunofluorescence assay in the BV2 cells under LPS treatment. The results indicated that P50 was higher in the nuclei of the BV2 cells stimulated by LPS than in the control group, and these increases were almost eliminated in the Ergosterol pretreatment cells in a concentration-dependent manner (Figure 2G). The above data indicated that Ergosterol could suppress inflammation activity by inhibiting the NF-κB signaling pathway in microglial cells stimulated by LPS.

### 2.3. Ergosterol Inhibits the Activation of MAPK and AKT Signaling Pathways Induced by LPS in Microglial Cells

Cytokine expression was also regulated by the MAPK and AKT signaling pathways. We detected the influences of Ergosterol on the activation of JNK and P38 kinase, two essential kinases in the MAPK pathway, in the BV2 and HMC3 cells treated with Ergosterol and LPS. Western blot detection displayed that LPS treatment increased the levels of p-JNK and p-P38 kinase, and Ergosterol pretreatment suppressed this increase in a dose-dependent manner in the BV2 (Figure 3A,C) and HMC3 cells (Figure 3B,D). We further examined the effect of Ergosterol on AKT signaling. Western blot detection showed similar results, with Ergosterol significantly inhibiting the increase in p-AKT levels induced by LPS in both types of microglial cells (*p* < 0.05, Figure 3A–D). In summary, these results suggest that Ergosterol could also suppress inflammation activity by inhibiting the MAPK and AKT signaling pathways in microglial cells.

### 2.4. Ergosterol Suppresses Neuroinflammation in LPS-Treated ICR Mice

As Ergosterol can inhibit microglial activation by targeting multi-pathways, we attempted to confirm its preventive function on neuroinflammation in vivo. As shown in Figure 4A, the ICR mice were first injected with Ergosterol (25 or 50 mg/kg, or vehicle intraperitoneally (i.p.)) daily for 5 days, and then, LPS (5 mg/kg, i.p.) or phosphate balanced solution (PBS) was injected. After 24 h, the mice were perfused and sacrificed, and the brain tissue was used to perform immunohistochemistry, qRT-PCR, and Western blot detection. The immunohistochemical results revealed that the expression of IBA-1 was higher in the hippocampus and cerebral cortex of the LPS-stimulated mice. However, Ergosterol dramatically reduced the IBA-1 protein level, especially in the group treated with high concentrations of Ergosterol (Figure 4B,C). qRT-PCR detection showed that a high concentration of Ergosterol treatment weakens the mRNA increase in IL-1β, interleukin-6 (IL-6), TNF-α, and INOS in the hippocampus and cortex of mice treated with LPS (Figure 4D,E). Western blot assay also displayed that Ergosterol pretreatment eliminated the protein levels of p-P65, IL-6, and IBA-1 enhanced by LPS (Figure 4F,G). Moreover, we determined the IL-1β and IL-6 protein levels in the plasma of mice using ELISA and found that Ergosterol could reduce the release of these cytokines induced by LPS. Finally, the expression level of NO, a neurotransmitter and cytotoxin produced by activated microglia, was ascertained in the plasma of mice. The ELISA results showed that Ergosterol markedly reduced the NO levels in LPS-treated mice. The above results illustrated that Ergosterol pretreatment could suppress neuroinflammation in vivo.

### 2.5. Ergosterol Reduces Neuron Damage in LPS-Treated Mice

Excessive neuroinflammation can cause neuron damage, resulting in a reduction in a great number of synapses. To evaluate the effect of Ergosterol on neuron damage, we checked the expression levels of protein involving the neural conduction of synapses, including postsynaptic density protein 95 (PSD95), Synapsin-1, and Synaptophysin, in both the hippocampus and cortex of mice. The Western blot results displayed that LPS treatment obviously reduced the PSD95, Synapsin-1, and Synaptophysin protein levels in the cortex, which were recovered in the Ergosterol pretreated group (Figure 5A,B). We subsequently detected Synapsin-1 by immunohistochemistry in both the hippocampus and cortex of mice, finding that LPS reduced the fluorescence intensity of Synapsin-1. At the same time, Ergosterol pretreatment significantly increased Synapsin-1 immunoreactivity, with the higher-concentration Ergosterol treatment showing this more clearly (Figure 5C,D). Above all, these results illustrated that Ergosterol pretreatment could clearly reduce the neuron damage induced by neuroinflammation.

## 3. Discussion

Neuroinflammation often has a great impact on the progression of central nervous system (CNS) injury, infection, toxicity, or autoimmunity [21]. The activation of microglia is the main characteristic of neuroinflammation, which promotes the release of neurotoxic molecules [22] and pro-inflammatory cytokines [23], ultimately leading to progressive neuronal cell degeneration [24]. Thus, inhibiting the overactivation of microglia is an attractive prospect for alleviating the inflammatory process of the nervous system.

To relieve neuroinflammation, and as a valuable source of new medicines for the treatment of nervous system diseases, natural compounds have received widespread attention. Astragaloside IV, one of the main active ingredients in *Radix Astragali*, can significantly reduce the LPS-induced inflammatory mediators such as NO, TNF-α, and IL-6 in BV2 and primary microglial cells [25]. It has been shown that sinomenine, an alkaloid isolated from the *Sinomenium acutum*, is recommended to patients with rheumatoid arthritis, to protect neurons in a model of PD in vitro and in vivo [26].

*Antrodia camphorata* is a rare and valuable medicinal fungus that lives on the endangered tree Camphor, which is endemic in Taiwan. There is a study that demonstrated the antioxidant, anti-inflammatory and anti-atherosclerotic properties of *Antrodia camphorata* [27]. It is reported that the methanol extract of *Antrodia camphorata* can efficiently inhibit the protein expression of COX-2 and INOS, the degradation of NF-κB and IκBα, and the phosphorylation of MAPKs; it also can suppress pro-inflammatory mediators such as NO, TNF-α, IL-1β, and IL-6 during LPS-induced lung injury [28]. The extract of *Antrodia camphorata* fruiting bodies showed significant neuroprotective activities in PC-12 cells [29]. These studies suggested that *Antrodia camphorate* has anti-inflammatory properties by interfering with multiple pathways.

The ingredients of *Antrodia camphorata* are complicated. It is reported that Ergostatrien-3β-ol (EK100) isolated from *Antrodia camphorata* can suppress oxidative stress and inflammation to protect the liver from hepatic ischemia/reperfusion injury [30]. Ergosterol isolated from cloud ear mushroom is reported to attenuate the inflammation in BV2 microglia cells [16,19]; however, the function in the nervous system was not investigated. In our study, we found that Ergosterol could inhibit cytokine expression in LPS-stimulated microglia to exert neuroprotective activity in vitro and in vivo. Although the above study reported that Ergosterol has a killing effect on BV2 cells at a concentration of 100 nM [19], in our experiments, Ergosterol had no obvious killing effect on HMC3 cells at a concentration of 100 μM and little effect on BV2 cells; thus, we used Ergosterol at concentrations of 10, 30, and 50 μM in further experiments. Another similar study investigated the effect of various concentrations of Ergosterol, including higher than 100 μM, on the viability of BV2 cells, and the results showed no significant difference in cell survival rate compared with a negative control [16]. These results suggest that different experimental environments and conditions may affect the impact of drugs; thus, we will study the toxicological safety of this drug in future studies.

LPS is used worldwide in experimental models of neuroinflammation both in vitro and in vivo [31]. The LPS animal model has several advantages, including technical ease and high reproducibility, especially in the activation of inflammation. Shortly after LPS administration, overproduction of pro-inflammatory cytokines in circulating serum can be detected [32]. Our study demonstrated that Ergosterol isolated from *Antrodia camphorata* had the strongest anti-inflammatory activity in two kinds of microglia cells treated with LPS through multiple signaling pathways. The results demonstrated that Ergosterol inhibited the LPS-induced phosphorylation of NF-κB P65 and the expression of NF-κB P50 in the nucleus in microglia cells. Furthermore, Ergosterol suppressed the phosphorylation of MAPKs and AKT in a concentration-dependent manner. The pathway of MAPK is widely present in the intracellular signal transduction of eukaryotic cells and is generally believed to include three members: JNK, extracellular-regulated protein kinases (ERK), and P38 kinase. When we established the LPS-induced inflammation model, we found that the levels of phosphorylated JNK and P38 increased, but the phosphorylated ERK (p-ERK) did not change under 1 μg/mL of LPS treatment. There is a report showing that the levels of p-ERK increased under the induction of LPS at 2 μg/mL on BV2 cells [33]. It is possible that p-ERK induction needs a higher concentration of LPS. Our findings illustrated that Ergosterol may target factors in multiple signaling pathways, which act together to inhibit inflammation in microglia cells.

As an ingredient of Gram-negative bacteria, LPS induces the production of pro-inflammatory mediators by targeting TLR4 [34]. Moreover, the NF-κB, MAPK, and AKT signaling pathways are downstream and can be regulated by TLR4. The inhibitory effect of Ergosterol on multiple signaling pathways leads us to believe that the upstream of these signaling pathways such as TLR4 might be closely related to the target of Ergosterol; we will further explore the detailed mechanism of anti-inflammation of Ergosterol in future studies. Therefore, it was important to determine the role of Ergosterol in the interaction between LPS and TLR4 in the regulation of neuroinflammatory responses.

Microglia cells participate in immunity monitoring and host defense under physiological conditions [35]. Under inflammation, however, the abnormal activation of microglia, manifested by the increased expression of the IBA-1 marker, may cause brain injury through the release of a variety of neurotoxic mediators, including reactive nitrogen species [36]. Significant evidence points out that activated microglia, mediated by INOS releasing NO, play a key role in the occurrence of neurodegenerative diseases, resulting in neuronal death via necrotic and apoptotic pathways [37]. Our study demonstrated that Ergosterol pretreatment conspicuously decreased the level of NO in LPS-treated mice and protected the neurons from injury. Moreover, the pro-inflammatory reaction, or the changes caused by activated microglia, in acute to chronic neuroinflammation is a fundamental process in the progression of several neurodegenerative diseases [4]. Synaptic degeneration is an early change in neurodegenerative diseases, and synaptic proteins are important components for maintaining the function of synapses. Synaptophysin is often used as a marker for quantifying the number of intact synapses [38]. Synapsin-1 is a presynaptic protein, and PSD95 is postsynaptic protein, both of which constitute the molecular basis for synaptic plasticity [39,40]. The expression of these proteins will decrease when neurons are injured. Our data showed that the levels of these proteins in the mice brain tissues treated with LPS after Ergosterol treatment were increased, which suggested that Ergosterol also had protective effects on neuron damage by suppressing the activation of microglia. In the future, it will be essential to evaluate the anti-inflammation activity of Ergosterol in neurodegenerative diseases, including PD and AD. Our data provide solid evidence for the potential application of Ergosterol as a drug in controlling neuroinflammation.

In conclusion, the present study revealed that Ergosterol could inhibit the expression of cytokines in LPS-induced microglia in vitro and in vivo. Further research found that Ergosterol exerted anti-inflammation activity by inhibiting the NF-κB, MAPK, and AKT signaling pathways. Moreover, Ergosterol could reduce the neuron damage induced by neuroinflammation in the hippocampus and cortex of mice. These findings suggest that Ergosterol has new therapeutic effects in the absence of neuroinflammation and neuroprotection from LPS-induced toxicity and may provide new leading compounds to tackle neuroinflammation-related diseases.

## 4. Materials and Methods

### 4.1. Cell Culture

The mouse microglial cell line BV2 and the human microglial cell line HMC3 were purchased from the Cell Bank of the Chinese Academy of Science (Shanghai, China). The BV2 and HMC3 cells were respectively grown in RPMI-1640 medium (Gibco, Logan, UT, USA) and Dulbecco’s modified Eagle medium (DMEM, Gibco, Logan, UT, USA). Two cell lines were cultured with 5% CO_2_ at 37 °C, and the culture medium was supplemented with 10% fetal bovine serum (FBS, Gibco, Logan, UT, USA).

### 4.2. Compounds

The natural compounds library contained 49 compounds and were purified from Fujian and Taiwan characteristic Chinese herbal medicine, including *Antrodia camphorata*, *Astragalus*, *Nasturtium*, *Vernonia amygdalina Delile*, *Hypericum elodeoides Choisy.* These traditional Chinese medicines have been used as anti-inflammatory drugs [41,42,43].

The purity of compounds was appropriate to 95% determined by HPLC (Agilent, Santa Clara, CA, USA). The anti-inflammation activity screening showed that four compounds (5, 24, 48, 49) from the natural compounds library possess bioactive effect on BV2 cell, and compound 5, which showed better anti-inflammatory effects, was chosen for further research. Meanwhile, the source, extraction, isolation and identification of compound 5 is shown as follows.

*Antrodia camphorata* (400 g) were chopped into pieces and then extracted with 100% ethanol at temperature three times (3 × 10 L, 12 h). The solution was concentrated to yield 104 g crude extract, which was subjected to silica gel chromatography and eluted with N-hexane-EtOAc (9:1 to 1:1) and CH_2_Cl_2_-MeOH (9:1 to 1:1) to give 16 fractions (A-P). Fraction G (0.9 g) was recrystallized to yield compound 5 (90 mg). The detail separation process of compound 5 is shown in Appendix A.

Compound 5, obtained as colorless acicular crystal, was assigned the molecular formula C_28_H_44_O by the analysis of its high resolution electrospray ionization mass spectroscopy (HRESIMS) with a sodium adduct ion at *m/z* 379.3360 [M-H_2_O+H]^+^ (calcd for C_28_H_43_) (Appendix A). In the ^1^H NMR spectrum (Appendix A and Appendix A), four olefinic proton signals at 5.50 (1 H, dd, *J* = 5.5, 2.2 Hz), 5.32, (1 H, m), 5.16 (1 H, dd, *J* = 15.3, 7.3 Hz), 5.10 (1 H, dd, *J* = 15.3, 7.3 Hz), one oxymethine proton signal at *δ*_H_ 3.57 (1 H, tt, *J* = 11.1, 4.1 Hz), and six methyl proton signals at *δ*_H_ 0.97, (3 H, d, *J* = 6.8 Hz), 0.88, (3 H, s), 0.85 (3 H, d, *J* = 6.8 Hz), 0.77 (3 H, d, *J* = 6.8 Hz), 0.76 (3 H, d, *J* = 6.8 Hz), 0.56, (3 H, s) were observed. The ^13^C NMR spectrum (Appendix A and Appendix A) exhibited 28 carbon signals, including six olefinic carbon at *δ*_C_ 141.4, 139.8, 135.6, 132.0, 119.6, 116.3, one oxymethine at *δ*_C_ 70.5, and six methyl at *δ*_C_ 21.1, 20.0, 19.7, 17.6, 16.3, 12.1. The NMR data of compound 5 were the same as that of Ergosterol, according to Reference [44]. Thus, the structure of compound 5 was assigned. The structures of compounds 24, 48 and 49 were defined as corymbocoumarin, Ua1 and Uf2, respectively, through NMR and HRESIMS data (Appendix A). NMR spectroscopic data were recorded using a Bruker AV-600 instrument. Mass spectrometry analysis was conducted on a Q Exactive Orbitrap LC-MS/MS (ThermoFisher, Waltham, MA, USA).

All compounds were dissolved as a 50 mM stock in DMSO (Thermo Fisher Scientific, Waltham, MA, USA).

### 4.3. Antibodies and Inhibitors

We used the following primary antibodies for Western blot and immunohistochemistry in Table 1.

SP600125, used as a JNK pathway inhibitor, was purchased from MedChe Express (HY-12041, Shanghai, China). SP600125 was prepared as stock solutions (50 mM) and diluted to a final concentration of 10 µM. LPS (from *E. coli* 055:B5) was purchased from Sigma (Sigma-Aldrich, China).

### 4.4. Animal and Treatment

All experiments were conducted according to approved animal protocols and guidelines set by the School of Medicine, Jianghan University (approval number: JHDXLL2022-073). Male ICR mice (6 weeks) weighing 30–35 g were purchased from Beijing Vital River Laboratory Animal Technology Co. Ltd. All mice were raised in a pathogen-free facility with a 12 light/12 dark cycle at a temperature of 23 ± 2 °C and fed with normal water and food. After a week of acclimation, 32 mice were divided into four groups (*n* = 8) randomly: the negative control group (Nacl), the LPS (5 mg/kg) group (model), the low-dose Ergosterol and LPS treatment group (ELT-L, 25 mg/kg), and the high-dose Ergosterol and LPS treatment group (ELT-H, 50 mg/kg). The ICR mice were injected daily with Ergosterol or vehicle (4% DMSO + 5% Tween 80) intraperitoneally (i.p.) for 4 days and were injected with LPS (5 mg/kg) or PBS on the fifth day. The mice were perfused and sacrificed for further research at 24 h after injection.

### 4.5. Immunohistochemistry

The NF-κB localization of BV2 cells was detected by immunofluorescence. The cells were cultivated overnight on sterile glass slides and fixed with 4% methanol. The samples were sealed with 5% BSA and 0.1% Tritonx-100, washed three times with 1 × PBS, and then incubated at 4 °C with rabbit polyclonal antibodies at 1:100. After rinsing with TBST, the IgG Alexa Fluor 488-conjugated anti-rabbit antibody (Invitrogen, Carlsbad, CA, USA) was incubated for 1 h at room temperature. The nuclei were stained with Hoechst33342 (Sigma Aldrich, St. Louis, MO, USA). A fluorescent microscope (BX51, Olympus) was used to capture images.

The brain tissues of the experimental mice were fixated by perfusion with PBS and 4% paraformaldehyde (PFA) solution and then dehydrated with 30% sucrose. The brain tissues were rapidly frozen and sectioned with a cryostat (30 μm thick). The brain slices were washed with PBS and infiltrated with PBS containing 0.5% Triton X-100 for 15 min, and then, the PBS containing 0.1% Triton X-100 and 5% BSA was sealed at room temperature for 1 h. Tissue sections were incubated with primary antibody (Table 1) at 4 °C overnight. Subsequently, the brain sections were washed three times with PBS and incubated with secondary goat IgG Alexa Fluor 488-conjugated anti-rabbit antibody (Invitrogen, Carlsbad, CA, USA) at room temperature for 1 h. The brain slices were then washed with PBS three times, installed on slides, and installation solution containing DAPI (P0131–25 mL, Beyotime Biotechnology, Shanghai, China) was applied. Images were acquired by confocal microscopy at ×20 (STELLARIS 5, Leica Microsystems, Wetzlar, Germany). The fluorescence intensity was quantified using ImageJ [45].

### 4.6. Cell Viability

The BV2 and HMC3 cells were seeded at 5 × 10^4^ cells/well in 96-well plates, separately, and treated with a series of concentrations of Ergosterol (0, 5, 10, 30, 50, 100 μM) for 24 h. A Cell Counting Kit-8 (CCK8, Beyotime Biotechnology, Shanghai, China) assay was used to evaluate the cell viability in accordance with the manufacturer’s instructions. Each well was filled with CCK-8 solution and incubated for 3 h at a temperature of 37 °C. A microplate reader (Bio-Rad, Hercules, CA, USA) was used to measure the absorbance at 450 nm. The cell viability was calculated as (the absorbance of the Ergosterol-added cells/the absorbance of the non-added cells) × 100%.

### 4.7. Quantitative Real-Time PCR Assay

Total RNA from cells or mice tissues was purified with the Trizol reagent (Invitrogen, Carlsbad, CA, USA) according to the manufacturer’s instructions. M-MLV Reverse Transcriptase (Takara Bio, Tokyo, Japan) was used for 2 μg RNA reverse transcription, and the cDNAs were used as templates for quantitative real-time PCR (qRT-PCR) amplification using TB Green^®^ Premix Ex Taq™ II (Takara Bio, Tokyo, Japan) to determine the mRNA expression levels. The qRT-PCR CFX 96 thermocycler (Bio-Rad Laboratories, Inc., Hercules, CA, USA) was used for the PCR reaction with 20 µL specific primers. The 2−ΔΔCt method was used to calculate the relative quantity of PCR products. The GAPDH gene was standardized. The primers for qRT-PCR are listed in Table 2.

### 4.8. Enzyme-Linked Immunosorbent Assay

The blood of the experimental mice was collected, and the plasma was centrifuged with 2000× *g* centrifugation at 4 °C for 20 min. A relevant ELISA (Bioswamp Life Science Lab, Wuhan, China) kit was employed to measure the level of IL-1β and IL-6 in plasma following the manufacturer’s instructions.

### 4.9. Western Blot

The cells were lysed with RIPA buffer supplemented with protease and phosphatase inhibitor (Beyotime, Shanghai, China), and the protein concentration of the samples was determined with a BCA kit (Beyotime Biotechnology, Beijing, China). Next, 10 µg of protein was separated using 10% sodium dodecyl sulfate-polyacrylamide gel electrophoresis (SDS-PAGE) and then transferred onto a polyvinylidene fluoride (PVDF) membrane (Millipore, Burlington, MA, USA). The membranes were sealed for 1 h with 5% non-fat milk, followed by incubation with primary antibodies (Table 1) at 4 °C overnight. The membranes were then incubated with suitable HRP-conjugated secondary antibodies (Boster Biological Technology, Wuhan, China) at an ambient temperature for 1 h. The bands were visualized with Chemidoc XRS Gel Imaging using an ECL Substrate (Millipore, Burlington, MA, USA). Finally, membranes were stripped and re-probed. Each band was quantified by densitometry using Image J, β-actin used for normalization [46].

### 4.10. Griess Assay (NO Assay)

The NO concentration in the plasma of the mice in the experiment was assayed according to the Griess reaction method (Beyotime Biotechnology, Shanghai, China). The absorbance was measured at 540 nm, and the NO content was analyzed with reference to the standard sodium nitrite curve.

### 4.11. Statistical Analysis

All data are expressed in mean and standard error (mean ± SEM). All tests were carried out in triplicate. GraphPad Prism 5 software (GraphPad Software, Inc., San Diego, CA, USA) was used with the analysis of variance after Dunnett’s test and Student’s two-tailed *t* test to calculate the statistical significance difference. The significance standard was set at *p* < 0.05.

## Figures and Tables

**Figure 1 molecules-28-02406-f001:**
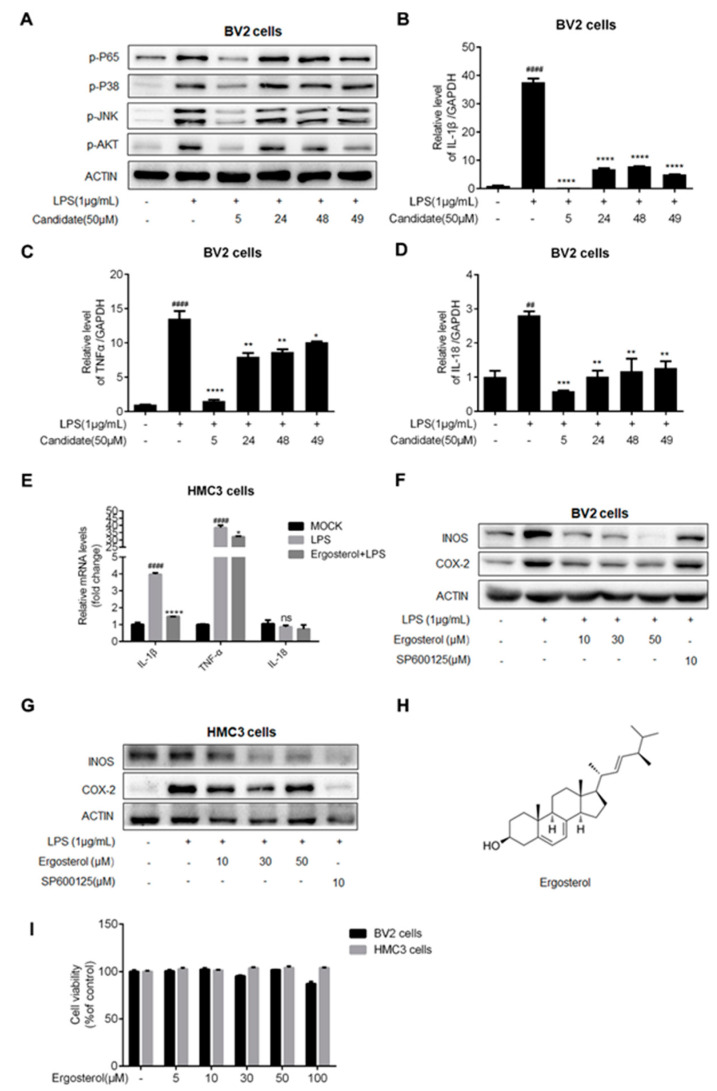
Ergosterol inhibits the microglial inflammation induced by LPS treatment. (**A**) BV2 cells pretreated with a natural compound for 2.5 h, and then LPS added at a final concentration of 1 μg/mL. After 30 min, the protein levels of p-P65, p-P38, p-JNK, and p-AKT were detected using Western blot. (**B**–**D**) BV2 cells were incubated with Ergosterol for 2.5 h, and then, LPS was added. After 6 h, the mRNA levels of IL-1β, TNF-α, and IL-18 were detected by qRT-PCR. (**E**) The same experimental procedure was performed in the HMC3 cells to detect the mRNA levels of IL-1β, TNF-α, and IL-18 by qRT-PCR. (**F**,**G**) BV2 and HMC3 cells pretreated with Ergosterol or SP600125 for 2.5 h, followed by LPS stimulation for 22 h; Western blot was used to determine the protein levels of INOS and COX-2. (**H**) Chemical name and structure of Ergosterol. (**I**) BV2 and HMC3 cells were seeded on 96-well plates and then treated with Ergosterol (0, 5, 10, 30, 50, 100 μM) for 24 h. Cell viability was assessed using a Cell Counting Kit-8 (CCK8) assay. All data are mean ± SEM of triplicate values. Statistical significance was analyzed using one-way ANOVA and *t* test. # Significantly different from the control; * significantly different from the LPS-treated group. Significance: ^##^ *p* < 0.01, ^####^ *p* < 0.0001, * *p* < 0.05, ** *p* < 0.01, *** *p* < 0.001, **** *p* < 0.0001.

**Figure 2 molecules-28-02406-f002:**
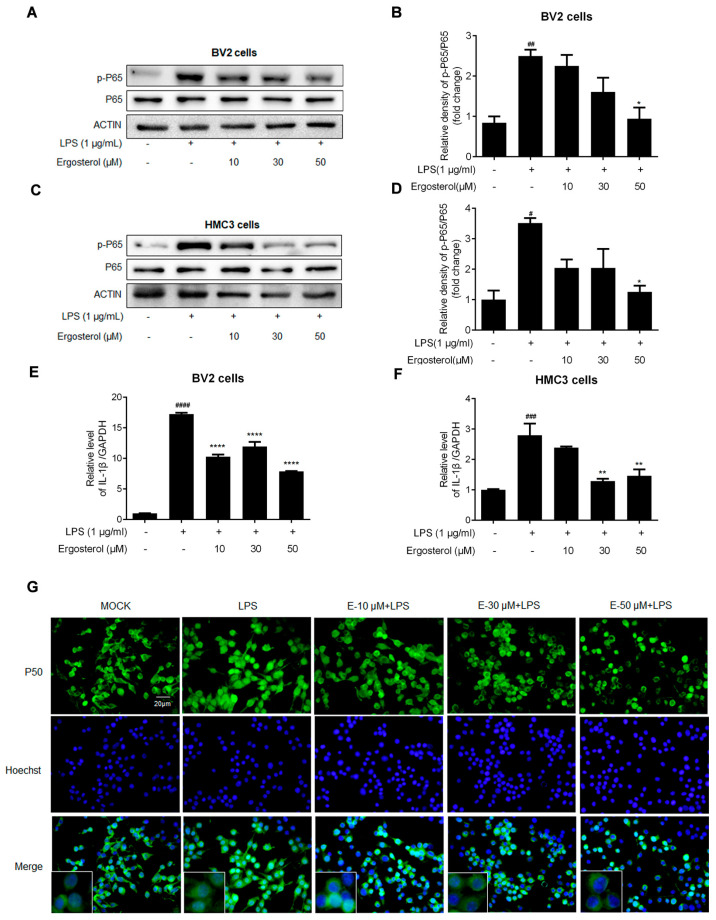
Ergosterol suppresses activation of the NF-κB signaling pathway in BV2 and HMC3 cells stimulated by LPS. BV2 and HMC3 cells pretreated with Ergosterol for 2.5 h, then LPS stimulated (1 μg/mL) for 30 min; the cells were collected and lysed for protein detection, or the cell RNA was purified for qRT-PCR determining. (**A**–**D**) The protein levels of P65 and p-P65 in cell lysates were detected by Western blot. Actin was used as a housekeeping control. (**E**,**F**) The mRNA levels of IL-1β were detected with qRT-PCR. (**G**) Detection of the expression and localization of NF-κB P50 (P50) in BV2 cells by immunostaining with an anti-P50 antibody (green) and Hoechst for nucleus staining (blue). The image was observed and recorded with a fluorescence microscope; scale bar is 20 μm. All data are mean ± SEM of triplicate values. Statistical significance was analyzed using one-way ANOVA and *t* test. # Significantly different from the control; * significantly different from the LPS-treated group. Significance: ^#^ *p* < 0.05, ^##^ *p* < 0.01, ^###^ *p* < 0.01, ^####^ *p* < 0.0001, * *p* < 0.05, ** *p* < 0.01, **** *p* < 0.0001.

**Figure 3 molecules-28-02406-f003:**
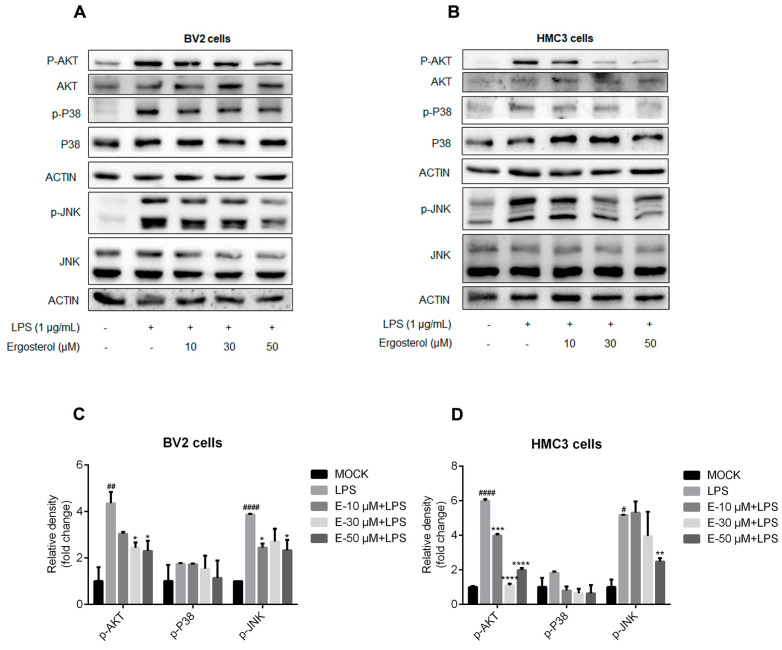
Ergosterol suppresses the activation of the MAPK and AKT signaling pathways in BV2 and HMC3 cells stimulated by LPS. (**A**,**C**) BV2 microglial cells were pretreated for 2.5 h at the indicated concentration of Ergosterol and then stimulated by LPS (1 μg/mL) for 30 min. The protein levels of P38, p-P38, JNK, p-JNK, AKT, and p-AKT were determined by Western blot in the BV2 cells. Actin was used as a housekeeping control. (**B**,**D**) Similar experiments were performed with the HMC3 cells. All data are mean ± SEM of triplicate values. Statistical significance was analyzed using one-way ANOVA and *t* test. # Significantly different from the control; * significantly different from the LPS-treated group. Significance: ^#^ *p* < 0.05, ^##^ *p* < 0.01, ^####^ *p* < 0.0001, * *p* < 0.05, ** *p* < 0.01, *** *p* < 0.001, **** *p* < 0.0001.

**Figure 4 molecules-28-02406-f004:**
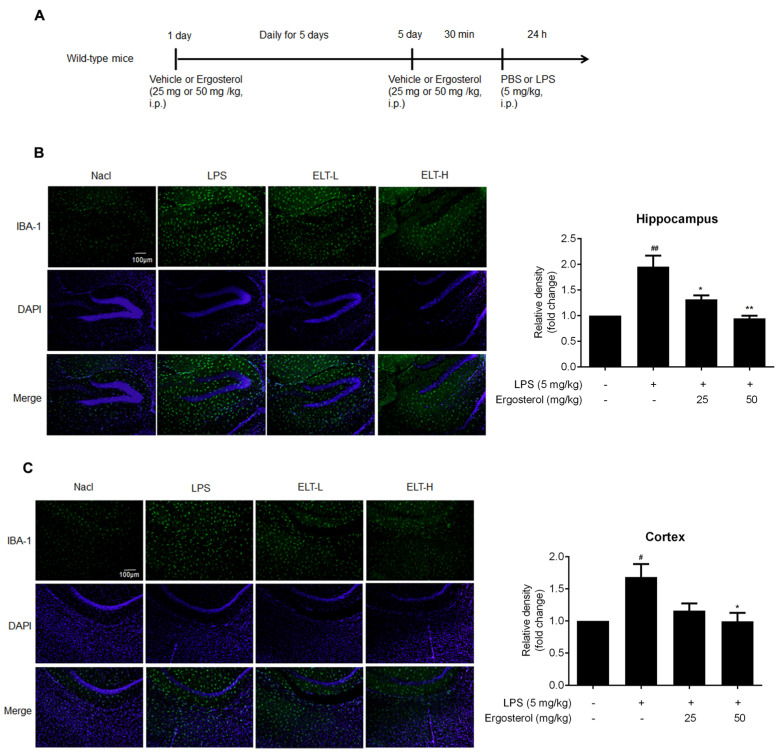
Ergosterol markedly decreased LPS-stimulated microglial activation and inflammation in ICR mice. (**A**) ICR mice were injected daily with Ergosterol (25 mg/kg, 50 mg/kg, i.p.) or vehicle (PBS, i.p.) for 5 days, followed by LPS (5 mg/kg, i.p.) or PBS injected for 24 h. The mice were then perfused and fixed. (**B**,**C**) An anti-IBA-1 antibody was used for immunohistochemistry. Quantification of IBA-1 fluorescence strength in the cerebral cortex or hippocampus (Nacl, *n* = 8 mice; LPS, *n* = 8 mice; Ergosterol 25 mg/kg + LPS is marked as ELT-L, *n* = 8 mice; Ergosterol 50 mg/kg + LPS is marked as ELT-H, *n* = 8 mice). (**D**,**E**) The mRNA levels of the IL-1β, IL-6, TNF-α, and INOS inflammatory factors in the cortex and hippocampus of the mice were measured with qRT-PCR. (**F**) The p-P65, IL-6, and IBA-1 protein levels were determined by Western blot. Actin was used as a housekeeping control (Nacl, *n* = 3 mice; LPS, *n* = 3 mice; ELT-L, *n* = 3 mice; ELT-H, *n* = 3 mice). (**G**) The relative expression level of protein to actin was calculated by densitometry. (**H**,**I**) The protein levels of IL-1β and IL-6 in plasma were detected by ELISA. (**J**) The expression levels of NO in plasma were detected with a Griess assay. Statistical significance was analyzed using one-way ANOVA and *t* test. # Significantly different from the control; * significantly different from the LPS-treated group. Significance: ^#^ *p* < 0.05, ^##^ *p* < 0.01, ^###^ *p* < 0.001, ^####^ *p* < 0.0001, * *p* < 0.05, ** *p* < 0.01, *** *p* < 0.001, and **** *p* < 0.0001.

**Figure 5 molecules-28-02406-f005:**
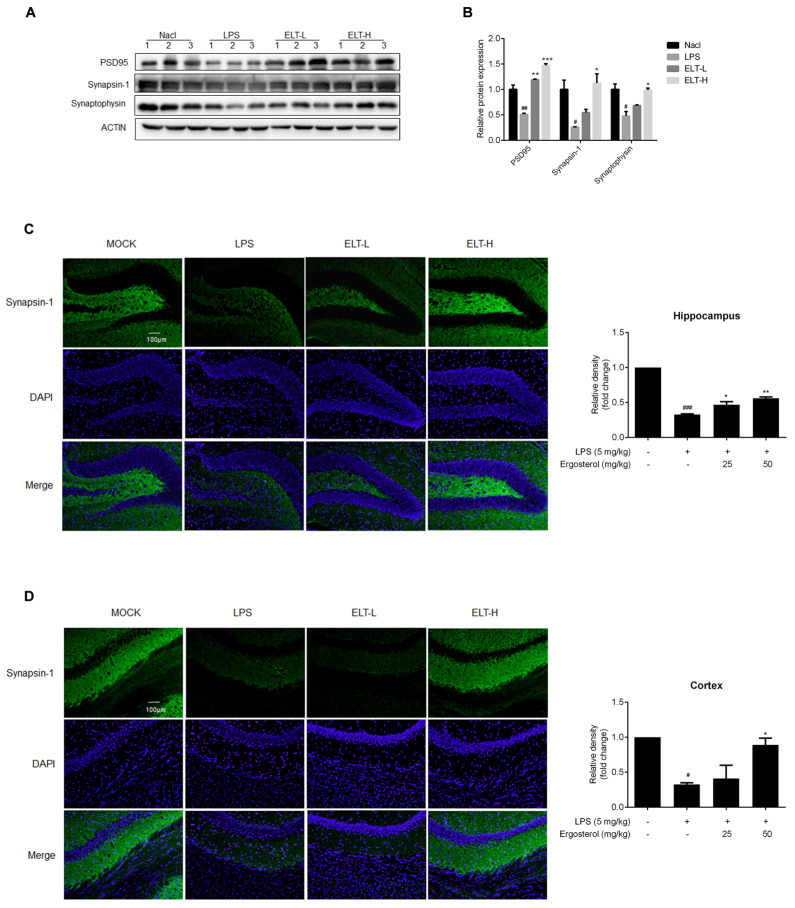
Ergosterol significantly reduced LPS-induced neuron injury in ICR mice. (**A**) The PSD95, Synapsin-1, and Synaptophysin protein levels in the cortex of mice were determined by Western blot. Actin was used as a housekeeping control (Nacl, *n* = 3 mice; LPS, *n* = 3 mice; ELT-L, *n* = 3 mice; ELT-H, *n* = 3 mice). (**B**) The densitometric method was used to estimate the relative expression of the protein compared to actin. (**C**,**D**) Immunohistochemistry was performed on anti-Synapsin-1 antibody. DAPI was used as a marker of the nucleus. Quantification of the relative fluorescence strength of Synapsin-1 in the region of the cortex or hippocampus was analyzed (Nacl, *n* = 8 mice; LPS, *n* = 8 mice; ELT-L, *n* = 8 mice; ELT-H, *n* = 8 mice). All data are mean ± SEM of triplicate values. Statistical significance was analyzed using one-way ANOVA and *t* test. # Significantly different from the control; * significantly different from the LPS-treated group. Significance: ^#^ *p* < 0.05, ^##^ *p* < 0.01, ^###^ *p* < 0.001, * *p* < 0.05, ** *p* < 0.01, and *** *p* < 0.001.

**Table 1 molecules-28-02406-t001:** List of antibodies used in this study.

Antibody	Manufacturer	Catalog No.
P50	Proteintech	14220-1-AP
P65	Cell Signaling	8242S
p-P65	Cell Signaling	3033S
AKT	Cell Signaling	9272S
p-AKT	Cell Signaling	4060S
P38	Cell Signaling	8690S
p-P38	Cell Signaling	4511S
JNK	Cell Signaling	9252S
p-JNK	Cell Signaling	9255S
INOS	Proteintech	22226-1-AP
Beta-actin	Proteintech	81115-1-RR
COX-2	Absin	abs131986
IL-6	Abcam	ab290735
IBA-1	Abcam	ab178846
Synapsin-1	Cell Signaling	5297S
Synaptophysin	Cell Signaling	25056S
PSD95	Cell Signaling	3450S

**Table 2 molecules-28-02406-t002:** The primers for qRT-PCR.

Primer	Forward (5′–3′)	Reverse (5′–3′)
mIL-1β	GCTGCTTCCAAACCTTTGAC	AGCTTCTCCACAGCCACAAT
mGAPDH	AGAACATCATCCCTGCATCC	CACATTGGGGGTAGGAACAC
mTNF-α	CCGATGGGTTGTACCTTGTC	CCGATGGGTTGTACCTTGTC
mIL-18	ACGTGTTCCAGGACACAACA	GGCGCATGTGTGCTAATCAT
mIL-6	CCACTTCACAAGTCGGAGGC	GGAGAGCATTGGAAATTGGGGT
mINOS	CCGGCAAACCCAAGGTCTAC	GCATTTCGCTGTCTCCCCAA
hIL-1β	ATGATGGCTTATTACAGTGGCAA	GTCGGAGATTCGTAGCTGGA
hGAPDH	CTGCACCACCAACTGCTT	TTCTGGGTGGCAGTGATG
hTNF-α	TATGGCTCAGGGTCCAACTC	GGAAAGCCCATTTGAGTCCT
hIL-6	ACTCACCTCTTCAGAACGAATTG	CCATCTTTGGAAGGTTCAGGTTG

## Data Availability

All generated and analyzed data used to support the findings of this study are included within the article.

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
