# Peer review of "Ergosterol Isolated from *Antrodia camphorata* Suppresses LPS-Induced Neuroinflammatory Responses in Microglia Cells and ICR Mice"

_molecules, 2023, doi:10.3390/molecules28052406_

Round 1
Reviewer 1 Report
The objective of this study is to show the suppressive effects of ergosterol on inflammatory responses of microglia cells in vitro and in vivo. Ergosterol was identified as an active compound among the natural compound library purified from Fujian and Taiwan traditional herbal medicine. The identification of ergosterol was done by NMR but it was not described any further. The anti-inflammatory effects of ergosterol were reported before but this study added new aspects to this suppressive function. The description of quantitative methods was inadequate (how to quantify signals of Western blots and histochemistry). Many Western blotting were only photos, and no quantitative analyses were presented. There are many descriptions that are not in standard English and are hard to understand. This reviewer highly recommends the authors seek an English editing service before manuscript submission. There are some abbreviations used without defining them first. This should be fixed. The followings are specific points to be considered.
L5 the affiliation of Jiazheng Guo is missing.
L44-61 many references, if not all, appear to be incorrect. Please check carefully.
L110 NMR method is not described.
Figure 1 How SP600125 was used should be described.
All figures description of * and # appears to be opposite.
L167 “significantly” this should be supported by actual statistical analysis presentation.
L247-248 Ergosterol is not corticosterol.
L277 “nitrogen” reactive nitrogen species?
L279 “poptotic” apoptotic?
L342 “a quarter” what is this?
Table 2 mNLRP3 was not used in the study.
Western blotting
Based on the method description and submitted original blotting photos, probing with an antibody was carried out using a separate membrane. The membranes were not reused by stripping the signals. If this is the case, what is the point to have beta-actin as an internal control?
L388 “nitrite oxide” nitric oxide?
Reviewer 2 Report
Comments:
In the present manuscript, Authors found the Ergosterol as an effective anti-neuroinflammatory agent by screening a natural compounds library. The mechanism was investigated both in vitro and in vivo. It could be reconsidered after major revision. Because
1, Where do these natural products come from, why are these natural products selected, and screening data are not presented or discussed in the text.
2, Importantly, the identification data of Ergosterol (No. 5 compound) as the key data is also not available in this manuscript. It should be presented in the manuscript.
3, Besides, ref.18 reported that more than 100 nM of Ergosterol significantly reduced the viability of BV2 cells while this manuscript showed that Ergosterol was safe in BV2 cell under 50 μM. It should be discussed in the text.
4, Although there are considerable data in this manuscript, there is a lack of in-depth analysis among these data. They need to be presented logically with discussion. Such as IBA -1, why only the activation of JNK and P38 were detected in the MAPK pathway? The manuscript should be re-edited.
5, There are numerous grammatical mistakes, because of which it is difficult to understand the message.
Round 2
Reviewer 1 Report
The revised manuscript is significantly improved. The authors responded well to the initial review to prepare for the revision. However, it is still not clear how the compound library was prepared and how compound 5 was identified as ergosterol by NMR. The supplement data show the NMR spectra of ergosterol, not compound 5. If the authors want to start this manuscript with traditional Chinese medicine, a more detailed description of a compound purification process should be included. Alternatively, the manuscript can be started with ergosterol effects without the section on Chinese medicine. Either way, the source of ergosterol should be listed. A reference for ImageJ should be included. L281-L283, mM should be µM.
Author Response
Response to Reviewer 1 Comments
Dear Reviewer 1:
Thank you very much for your patience and carefulness on our work. We are very grateful you for your comments and valuable advices. It was very helpful for improving the quality of our paper. We have revised our manuscript according to the advices. The following is our responses to your comments.
Point 1. It is still not clear how the compound library was prepared and how compound 5 was identified as ergosterol by NMR.
Response 1: We have added a preparation method for the compound library (Line 370-381 in Page 13) and displayed the experimental procedures at Figure S1. We analyzed four selected compounds with good anti-inflammatory effect in the screening experiment by high resolution electrospray ionization mass spectroscopy (HRESIMS) and NMR. We also added NMR detection detailed data and HRESIMS detection data in the section of materials and methods (Line 382-393 in Page 13). And the NMR diagram of compound 5 (Ergosterol) is attached as Table S1 and Figure S2-3. The HRESIMS diagram of compound 5 (Ergosterol) is attached as Figure S5. The structural assignment was accomplished by NMR and HRESIMS analyses, as well as comparing with the published data.
Point 2. The supplement data show the NMR spectra of ergosterol, not compound 5.
Response 2: We have changed the supplement legend (Figure S2-3) and the corresponding description is in the materials and methods (Line 392 in Page 13).
Point 3. If the authors want to start this manuscript with traditional Chinese medicine, a more detailed description of a compound purification process should be included. Alternatively, the manuscript can be started with ergosterol effects without the section on Chinese medicine. Either way, the source of ergosterol should be listed.
Response 3: We have added a more detailed description of a compound purification process and the source of Ergosterol in the section of materials and methods (Line 376-381 in Page 13).
Point 4. A reference for ImageJ should be included.
Response 4: We added the references (No.46-47) for ImageJ in the section of materials and methods (Line 442 in Page 15, Line 480 in Page 16).
Point 5. L281-L283, mM should be µM.
Response 5: We apologize for our mistake. We have corrected it in the revised manuscript (Line 292-294 in Page 11).
Reviewer 2 Report
The revised manuscript has been improved by authors. However, the several critical issues remain.
Major concerns:
1, The NMR data should be described in detail.
2, NMR data alone are not sufficient to identify the chemical structure of natural source ingredient. Other data or information should be added for its identification.
3, Although the authors have added discussion, it is still a low level of discussion that pulls down the overall level of the paper and does not meet the standards of Molecules. It is recommended that the discussion be focused on what they detected in the manuscript in conjunction with theory or existing literatures to enhance the logic and the scientific soundness of the manuscript.
Minor
3, Several mistakes are still existed. Such as lines 281-283, “mM” might be “μM”.
Author Response
Response to Reviewer 2 Comments
Dear Reviewer 2:
Thank you very much for your patience and carefulness on our work. We are very grateful you for your comments and valuable advices. It was very helpful for improving the quality of our paper. We have revised our manuscript according to the advices. The following is our responses to your comments.
Point 1. The NMR data should be described in detail.
Response 1: We have made a detailed description for the NMR data in the section of materials and methods (Line 385-392 in Page 13). And the NMR data of Ergosterol is attached as Table S1 and Figure S2-3.
Point 2. NMR data alone are not sufficient to identify the chemical structure of natural source ingredient. Other data or information should be added for its identification.
Response 2: We also performed high resolution electrospray ionization mass spectroscopy (HRESIMS) analysis to identify the chemical structure of 4 main compounds including Ergosterol. We added the HRESIMS data in the revised manuscript as Figure S5, 9, 13, 17, and inserted the corresponding description in the section of materials and methods (Line 382-384, 394-395 in Page 13).
Point 3. Although the authors have added discussion, it is still a low level of discussion that pulls down the overall level of the paper and does not meet the standards of Molecules. It is recommended that the discussion be focused on what they detected in the manuscript in conjunction with theory or existing literatures to enhance the logic and the scientific soundness of the manuscript.
Response 3: We have re-edited our discussion to enhance the logic and the scientific soundness of the manuscript according to your suggestion (Line 273-324, 346-348 in Page 11-12).
Point 4. Several mistakes are still existed. Such as lines 281-283, “mM” might be “μM”.
Response 4: We apologize for our mistake. We have corrected it in the revised manuscript (Line 292-294 in Page 11).